# *Campylobacter* spp. Prevalence in Santiago, Chile: A Study Based on Molecular Detection in Clinical Stool Samples from 2014 to 2019

**DOI:** 10.3390/pathogens12030504

**Published:** 2023-03-22

**Authors:** Lorena Porte, Caricia Pérez, Mario Barbé, Carmen Varela, Valeska Vollrath, Paulette Legarraga, Thomas Weitzel

**Affiliations:** 1Laboratorio Clínico, Clínica Alemana, Facultad de Medicina Clínica Alemana, Universidad del Desarrollo, Santiago 7650568, Chile; 2Facultad de Medicina Clínica Alemana, Instituto de Ciencias e Innovación en Medicina (ICIM), Universidad del Desarrollo, Santiago 7610507, Chile

**Keywords:** campylobacteriosis, epidemiology, gastrointestinal multiplex panel, PCR, diagnosis, South America

## Abstract

*Campylobacter* spp. is an emerging cause of infectious diarrhea worldwide. In South American countries such as Chile, its prevalence is underestimated due to inadequate detection methods. Gastrointestinal multiplex PCR panels (GMP) permit rapid and sensitive detection of bacterial pathogens and provide important epidemiological information. This study aimed to analyze *Campylobacter* epidemiology using the results of molecular methods and to compare molecular detection results to those of culture methods. We performed a retrospective, descriptive analysis of *Campylobacter* spp. detected in clinical stool samples between 2014–2019 by GMP and culture. Within 16,582 specimens examined by GMP, *Campylobacter* was the most prevalent enteropathogenic bacteria (8.5%), followed by *Salmonella* spp. (3.9%), *Shigella* spp./enteroinvasive *Escherichia coli* (EIEC) (1.9%), and *Yersinia enterocolitica* (0.8%). The highest *Campylobacter* prevalence occurred in 2014/2015. Campylobacteriosis affected more males (57.2%) and adults from 19–65 years (47.9%) and showed a bimodal seasonality with summer and winter peaks. In 11,251 routine stool cultures, *Campylobacter* spp. was detected in 4.6%, mostly *C. jejuni* (89.6%). Among 4533 samples tested by GMP and culture in parallel, GMP showed a superior sensitivity (99.1% versus 50%, respectively). The study suggests that *Campylobacter* spp. is the most frequent bacterial enteropathogen in Chile.

## 1. Introduction

*Campylobacter* spp. are emerging zoonotic pathogens inhabiting the gastrointestinal tract of different warm-blooded animals [1,2]. Human campylobacteriosis is endemic worldwide and usually manifests as acute diarrhea. *Campylobacter jejuni* is the most prevalent human pathogen. Campylobacteriosis has been the most frequent gastrointestinal infection reported in Europe during the last decade, with an incidence of 59.7 cases per 100,000 persons in 2019 [3,4]. Similar results have been reported from the USA, with an incidence of 19.5/100,000 during the same period [3].

Stool culture is the diagnostic standard for *Campylobacter* spp.; however, the high costs and complexity of this method have prevented its routine implementation in many resource-limited countries, although the pathogen seems to be of emerging epidemiological relevance in these regions [5,6]. The epidemiology and spectrum of campylobacteriosis in South America are uncertain [7]. Studies of different populations using various techniques found a prevalence ranging from 2.2% in Colombian children to 30.1% in children from Argentina [7]. Epidemiological studies in Chile demonstrated prevalence rates from 0.4%, using microscopy, to 18%, using culture [8]. A clinical study showed an increase in the detection of campylobacteriosis from 0.4% to 6.1% after the implementation of culture methods within routine stool workflow [5].

Gastrointestinal multiplex panels (GMP) based on polymerase chain reaction (PCR) have recently emerged as a sensitive, user-friendly, and rapid alternative to conventional culture methods [9]. This study aimed to analyze *Campylobacter* prevalence rates after the introduction of GMP and to compare molecular test performance to traditional culture methods.

## 2. Materials and Methods

We conducted a retrospective, descriptive study at the Clinical Laboratory of Clínica Alemana, a private nonprofit hospital in Santiago, Chile, between January 2014 and December 2019. The health center is located in an upper-income neighborhood and mainly serves this segment of the Metropolitan Region. Information was extracted from the clinical laboratory databases Sisalud (SONDA, Santiago, Chile) and KernMIC (bioMérieux, Marcy-l’Étoile, France). The diagnostic tests were applied according to the sender’s request. The data included demographics and the results of routine detection of *Campylobacter* spp. and other enteric bacterial pathogens (*Salmonella*, *Shigella,* and *Yersinia enterocolitica*) by two GMP, FilmArray gastrointestinal panel^®^ (bioMérieux, version 2.3) and A.I.I. Screen (Sacace Biotechnologies, Como, Italy). The former detects 13 bacterial, five viral, and four parasitic pathogens and was used from 2015–2019; the latter detects three bacterial (*Campylobacter*, *Salmonella*, and *Shigella*) and four viral enteropathogens, and was offered from 2014–2019. For their genetic similarity, *Shigella* spp. and enteroinvasive *Escherichia coli* (EIEC) were not distinguishable with either GMP. Assays were performed and interpreted according to the manufacturers’ instructions. If a pathogen was detected in the same patient more than once within a 30-day interval, only the first result was considered. The GMP detection rates of *Campylobacter* spp. and the three main bacterial enteropathogens (*Salmonella* spp., *Shigella* spp./EIEC, and *Yersinia enterocolitica*) were compared. Samples consisted of fresh stool, transported at ambient temperature, and processed within 2 h after collection.

In addition, the results of *Campylobacter* stool cultures performed during the study period were analyzed. Culturing was performed on CASA agar (bioMérieux, Marcy-l’Étoile, France) and plates were incubated at 42 °C under microaerobic conditions (Anaerocult^®^ C, Merck, Darmstadt, Germany) for 48 h. Quality control for each batch of culture medium was performed using *C. jejuni* ATCC 33291 (growth at 48 h) and *E. coli* ATCC 25922 (growth inhibition). *Campylobacter* isolates were confirmed and identified by MALDI-TOF mass spectrometry (Vitek MS, bioMérieux), as described previously [8]. Samples in which both GMP and culture were performed were used to analyze the diagnostic performance (sensitivity and specificity) of each method against two reference standards: (A) culture (true positive = culture positive) and (B) a composite standard (true positive = culture and/or GMP positive).

Analyse-it software (5.66.0 for MS Excel 10) was used for the statistical analysis and GraphPad version 9 (October 2020) for graph design.

## 3. Results

### 3.1. Gastrointestinal Multiplex Panels (GMP)

During the study period, a total of 16,582 GMP exams were performed and analyzed; the majority (11,278; 68%) by FilmArray. *Campylobacter* spp. was detected in 1412/16,582 samples (8.5%), while *Salmonella* spp., *Shigella* spp./EIEC, and *Yersinia enterocolitica* were identified in 653/16,582 (3.9%), 323/16,582 (1.9%), and 95/11,274 (0.8%) specimens, respectively. *Campylobacter* detection rates were highest in 2015 (12.6%) and subsequently declined, reaching their lowest rate in 2019 (4.8%) (Table 1). In contrast, the prevalence of *Salmonella* spp., *Shigella* spp./EIEC, and *Yersinia enterocolitica* remained stable throughout the study period.

Campylobacteriosis had a bimodal distribution with peaks during the summer months of January/February and winter months of July/August; such a winter peak was not observed with other bacteria (Figure 1).

The study population had an equal sex distribution (49.3% female; 50.7% male). *Campylobacter* spp. was significantly more frequent in male patients (57.2%). All bacterial pathogens analyzed were most frequently detected in the adult age group (19–65 years). Campylobacteriosis cases had a median age of 20 years (range 0–99), which was higher than *Salmonella* spp., but lower than *Shigella* spp./EIEC and *Yersinia enterocolitica* spp.; *Campylobacter* spp. was less frequent in children aged 0–5 years than *Salmonella* spp. (Table 2).

### 3.2. Campylobacter Culture

The overall prevalence of *Campylobacter* spp. among 11,251 routine stool cultures performed during the study period was 4.6% (CI95% 4.2–5.0). Among the 519 positive cultures, 465 (89.6%) were *C. jejuni* and 54 (10.4%) were *C. coli*.

We identified 4533 specimens for which physicians had ordered FilmArray GMP and *Campylobacter* culture from the same sample. Within this sample subset, 623 specimens (13.7%) were *Campylobacter*-positive; 313 (50.2%) were positive by both methods, 307 (49.3%) were positive only by GMP, and three samples (0.5%) were positive only by culture (Table 3).

Compared to culture as a reference standard, FilmArray GMP had a sensitivity of 99.1% (CI95% 97.2–99.7), a specificity of 92.7% (CI95% 91.9–93.5), a diagnostic accuracy of 93.2%, and a kappa coefficient of 0.6. The comparison of both methods to a composite standard showed that FilmArray GMP exhibited significantly higher sensitivity (99.5%; 98.6–99.8) than culture (50.7%; 46.6–54.4) (Table 4).

## 4. Discussion

In our analysis based on molecular detection methods, *Campylobacter* had a prevalence of 8.5% (CI95% 4.6–8.1) in clinical stool samples and represented the most common bacterial enteropathogen. *Salmonella* was second with a detection rate of 3.9%. To our knowledge, the present data set is the largest on human campylobacteriosis detected by molecular methods in South America. Results of deeper molecular analysis by whole genome sequencing of some of our strains were reported separately [10]. *C. jejuni* predominated, in accordance with previous local and global data [5,11].

In Chile, the true burden of campylobacteriosis is unknown, most probably reflecting the insufficient availability of diagnostic techniques [8,12]. Consequently, there is a lack of surveillance data and most available information has been generated within research projects with limited sample numbers. Between 2000 and 2019, Chilean studies using different culture strategies and/or rapid antigen detection tests reported a prevalence of campylobacteriosis ranging from 2.1% to 18% [5]. In 2013, an analysis in southern Chile using an in-house PCR in 140 patients with diarrhea found *C. jejuni* in 10.7% of cases [13]. Our study detected a prevalence of 8.5% using GMP, which is consistent with other GMP-based reports from Santiago, with rates between 7.3% and 12.7% [14,15].

Epidemiological information from other countries in South America is mainly based on different techniques for the culture and identification of *Campylobacter*, resulting in a wide range of prevalence rates (reviewed in [7]). For example, rates from Argentina varied from 9.1% in 2003, in a study performed in a low-income population, to 30.1% in 2010 in children younger than 15 years. In 2001, Bolivia reported a 4.4% frequency of detection for *C. jejuni* and a 7.3% frequency of detection for *C. coli* in patients with diarrhea. In 2002, a study from Peru found a prevalence of 13% for *C. jejuni* in infants between 0–2 years; in 2003, another study found *C. coli* (5%) to be more frequent than *C. jejuni* (2.9%) in patients with gastroenteritis. In 2007 and 2010, Paraguay and Uruguay, respectively, reported high rates of detection in children of 18.4% and 14.3%. In 2003, using a filtration culture technique, a study from Venezuela found a prevalence of 6.5% for *C. jejuni* in stool samples [7]. Colombia reported a low rate of campylobacteriosis in preschool children (0–5 years) with 2.3% in 2006 and 3.5% in 2013–2014 [7,11]. In 2010, Brazilian children with diarrhea had a prevalence of 9.6% based on a PCR method of detection [7]. These results are difficult to interpret due to the variability in methods, socio-economical settings, and age distributions; however, Campylobacter is likely an important gastrointestinal pathogen in most countries of the region.

The prevalence of *Campylobacter* showed significant differences over time, exhibiting a peak in 2015 followed by a decrease from 2016 to 2019. The reasons for this decline are unknown, since no other epidemiological information, e.g., outbreaks, is available for this period in Chile [16]. A bias due to the initial tendency to use GMP in more severe cases is unlikely since the prevalence of other organisms remained stable.

Interestingly, the seasonal distribution of *Campylobacter* spp. was bimodal, with peaks in summer (January–February) and winter (July–August). An increased prevalence during summer months has been reported worldwide [2,11,17]. Some studies have tried to identify the drivers of this seasonality, including improved survival and replication of microorganisms under warmer temperatures, transmission by flies, variations in animal colonization, seasonal changes in eating behavior and human recreational activities, and travel to endemic regions [11,17,18]. However, the mechanisms for the association between campylobacteriosis and warmer climate are still unclear [18,19]. Distinct but smaller annual winter peaks have been observed in some European countries (Austria, Belgium, Finland, Germany, Luxembourg, the Netherlands, Switzerland, and Sweden) and China (Beijing) [2,11]. Food habits or cooking trends could be a possible explanation. In Chile, until now, seasonal trends have not been reported. The observed summer and winter peaks, however, suggest that food preparation practices rather than environmental factors, such as temperature, might be more relevant.

We found a predominance of infection in males, which is similar to previous reports from Chile [20] and industrialized countries [11,21,22]. Louis et al. found that males represented 53.7% of campylobacteriosis cases in England and Wales between 1990 and 1999 in all age groups and all regions, suggesting that this might be related to a higher sex-specific susceptibility to infection [23]. Interestingly, a study from 2000–2003, in the same region, found a higher incidence of campylobacteriosis in males from birth to 17 years old, especially in the 13 to 15 years group. The authors postulated a relationship between hormonal changes during puberty in boys with increased growth of *Campylobacter* in the intestine. However, the male gender also predominated in patients 50 years and older, disempowering the hypothesis [24]. In a review of the global epidemiology of *Campylobacter* from 2014 to 2021, the infection was also more commonly found in males in most countries, with few exceptions [11]. A recent meta-analysis including national data from seven countries found that the incidence rates for campylobacteriosis were higher in males throughout all age groups, especially during puberty and in senior age. The authors speculated on a combination of multiple factors. In the older age group, men could be more prone to eat outside the home, getting exposed to undercooked meat, which, combined with a more common use of proton pump inhibitors at that age, could increase their risk of infection. They also referred to animal studies demonstrating that estrogen has a protective role against some enteropathogens and influences immune response through the gut microbiota [25].

Our data revealed a higher prevalence of *Campylobacter* in the adult group between 19 and 65 years old with a median age in cases of 20 years. This age distribution is in accordance with observations from several industrialized countries [4,11,26]. Similar trends have been found in urban communities from developing countries [11]. In developing countries, symptomatic infections usually affect children under 4 years of age, with the highest incidence in infants younger than 1 year [2]. In these settings, the frequency of clinical diseases caused by *Campylobacter* infections is assumed to decrease with age, reflecting the acquisition of immunity as a consequence of repeated exposure [27].

In addition, we used our databases to compare the performance of GMP and *Campylobacter* culture, using results from over 4500 samples, which were examined in parallel by both methods. As in the analysis of all samples, FilmArray GMP showed a higher *Campylobacter* prevalence than culture in this sample subset (13.7% versus 6.7%) [28]. Older studies evaluating molecular methods to detect *Campylobacter* in stool samples have used culture as a reference method [3]. Using this approach in our population, GMP showed a very high sensitivity (99.9%), but reduced specificity (92.7%). However, the interpretation of PCR-positive/culture-negative results is controversial [29,30,31]. On one side, molecular techniques detect *Campylobacter* DNA (not viable bacteria), which could lead to false-positive results. On the other hand, culture is limited by the fastidious nature of *Campylobacter*, possible antibiotic pretreatment, and the inability to detect species such as *C. upsaliensis*, which are inhibited by antibiotic supplements in culture media, leading to false-negative results [31]. More recently, PCR has been applied as a reference standard in some studies, resulting in a low sensitivity of culture methods [13,28,32]. Due to the absence of an accepted reference method, we applied a composite reference standard to evaluate the sensitivity of both methods. Using this approach, GMP displayed very high sensitivity (99.5%; CI95% 98.6–99.8) versus culture (50.7%; CI95% 46.8–54.6). A large North American prospective study comparing culture to molecular methods showed similar findings and concluded that PCR tests should have a major role in diagnostic testing for *Campylobacter* [31]. Despite the higher diagnostic sensitivity, GMP might not replace traditional culture methods, since they allow species identification and antimicrobial susceptibility testing. In addition, multianalyte PCRs often produce polymicrobial results of uncertain clinical relevance [9]. A preliminary analysis in our laboratory, however, showed that *Campylobacter*-positive samples had a lower co-infection rate than most other intestinal pathogens [33].

The study is limited by its retrospective nature and lack of clinical data. As a monocentric study in a high-income setting in Santiago, the results might not represent the Chilean population. Further multicentric studies are needed to confirm these findings.

## 5. Conclusions

This study provides important information on the epidemiological profile of campylobacteriosis in our population such as age groups, sex, and seasonality. The use of commercial gastrointestinal multiplex panels led to a significantly higher detection rate than culture, confirming the clinical importance of *Campylobacter* spp. as a highly prevalent bacterial enteropathogen in Chile.

## Figures and Tables

**Figure 1 pathogens-12-00504-f001:**
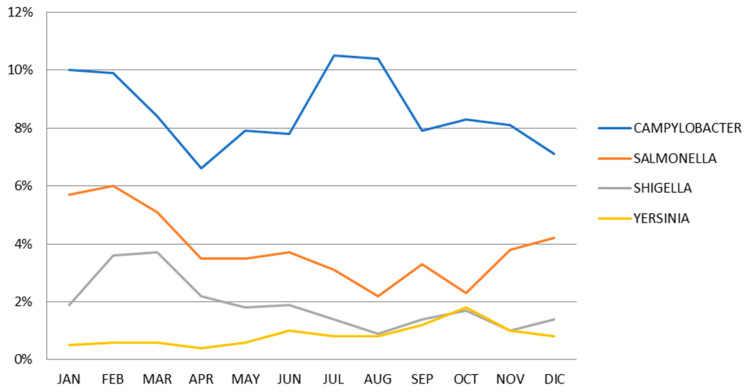
Seasonality (prevalence by month) of *Campylobacter* spp., *Salmonella* spp., *Shigella* spp./EIEC, and *Yersinia enterocolitica* detected using gastrointestinal multiplex panels (GMP) from 2014 to 2019.

**Table 1 pathogens-12-00504-t001:** Annual prevalence of *Campylobacter* spp. compared to other bacterial enteropathogens, as detected by gastrointestinal multiplex panels (GMP).

Year	*N*	*Campylobacter* spp.		*Salmonella* spp.		*Shigella* spp./EIEC		*Yersinia enterocolitica* *		
		Positive (%)	CI95%	Positive (%)	CI95%	Positive (%)	CI95%	*n* *	Positive (%)	CI95%
2014	1440	168 (11.7)	10.1–13.4	51 (3.5)	2.6–4.6	10 (0.7)	0.3–1.3	0		
2015	2279	285 (12.6)	11.2–13.9	103 (4.5)	3.7–5.5	43 (1.8)	1.3–2.5	489	5 (1.0)	0.3–2.4
2016	3238	296 (9.1)	8.0–10.2	125 (3.9)	3.2–4.6	53 (1.6)	1.2–2.1	2319	19 (0.8)	0.5–1.3
2017	3240	277 (8.5)	7.6–9.6	140 (4.3)	3.6–5.1	77 (2.4)	1.9–3.0	2768	26 (0.9)	0.6–1.4
2018	3124	230 (7.4)	6.5–8.3	117 (3.7)	3.1–4.5	76 (2.4)	1.9–3.0	2697	29 (1.1)	0.7–1.5
2019	3261	156 (4.8)	4.1–5.6	117 (3.6)	3.0–4.3	65 (2.0)	1.5–2.5	3001	16 (0.5)	0.3–0.9
Total	16,582	1412 (8.5)	8.1–9.0	653 (3.9)	3.7–4.3	324 (1.9)	1.7–2.2	11274	95 (0.8)	0.7–1.0

* *Yersinia enterocolitica* was only detected by FilmArray GMP.

**Table 2 pathogens-12-00504-t002:** Demographic data of 2483 patients infected with *Campylobacter* spp. compared to other bacterial enteropathogens, as detected by gastrointestinal multiplex panels (GMP).

Characteristic		*Campylobacter* spp. *n* = 1412		*Salmonella* spp. *n* = 653		*Shigella* spp./EIEC *n* = 323		*Yersinia enterocolitica* *n* = 95		Total *n* = 2483	
		*N* (%)	CI 95%	*N* (%)	CI 95%	*N* (%)	CI 95%	*N* (%)	CI 95%	*N* (%)	CI 95%
Sex	Female	604 (42.8)	40.2–45.4	308 (47.2)	43.3–51.1	163 (50.5)	44.9–56.0	42 (44.2)	34.0–47.0	1117 (45)	43–47
	Male	808 (57.2)	54.6–59.8	345 (52.8)	48.9–56.7	160 (49.5)	44.0–55.1	53 (55.8)	45.2–66.0	1366 (55)	53–57
Age (years)	Median	20		15		28		25		21	
	IQR	7–32		4–36		17–42		10–50		7–35	
Age groups (years)	0–5	312 (22.1)		201 (30.8)		33 (10.2)		17 (17.9)		563	
	6–18	348 (24.6)		151 (23.1)		51 (15.8)		23 (24.2)		573	
	19–65	674 (47.7)		267 (41.0)		221 (68.4)		44 (46.3)		1208	
	>65	77 (5.5)		34 (5.2)		18 (5.6)		11 (11.6)		139	

**Table 3 pathogens-12-00504-t003:** Results of *Campylobacter* detection by culture and FilmArray gastrointestinal multiplex panel (GMP).

		*Campylobacter* Culture		
		POS	NEG	Total
Filmarray GMP	POS	313	307	620
	NEG	3	3910	3913
	Total	316	4217	4533

**Table 4 pathogens-12-00504-t004:** Sensitivity of *Campylobacter* culture and FilmArray gastrointestinal multiplex panel (GMP) for the detection of *Campylobacter* spp. using a composite reference standard.

	Reference Standard A *			Reference Standard B **		
Method	True (+)	False (−)	Sens	True (+)	False (−)	Sens
Culture	316	0	100%	316	307	50.7%
FilmArray GMP	313	3	99.1%	620	3	99.5%

Sens = sensitivity; * True (+): samples positive by culture; ** True (+): samples positive by culture and/or FilmArray GMP.

## Data Availability

All relevant data supporting our findings are contained within the manuscript.

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
