# Peer review of "Campylobacter spp. Prevalence in Santiago, Chile: A Study Based on Molecular Detection in Clinical Stool Samples from 2014 to 2019"

_pathogens, 2023, doi:10.3390/pathogens12030504_

Round 1

Reviewer 1 Report

This an interesting, well designed study that provides epidemiological information about Campylobacter with data obtained over an extendedlong period of time and a large number of samples. Due to type of clinic where the study was carried out, it is inferred that the population was of better socioeconomic levels than the population consulting in the public health system. It should be necessary to explain this in the text. Probably the isolation rate could be assoicate to this factor (reference 15 shows higher frecuencies for lower income people). For the same reason,  it would not convenient to say that Campylobacter is the most frequent/prevalent enteropathogen in Chile (lines 26-27 and 250-251). The suggestion is to change to Campylobacter is the most frequent/prevalent enteropathogen in this series of Chilean patients. 

Reviewer 2 Report

The study by Porte at al. presents results that contribute to understanding the importance of Campylobacter spp. in Chile, a South American country where its prevalence is generally poorly known and underestimated. The manuscript is well written and presents relevant information. However, before being considered for publication by Pathogens (MDPI), authors should address the following issues:

In my opinion, the authors are mainly highlighting the advantages of GMP used (eg. high sensitivity). However, the disadvantages are not discussed: for example, the fact of not specifically knowing the Campylobacter species by GMP detection and more importantly, antibiotic resistance cannot be determined to make a more appropriate treatment.

What was the proportion of cases in which Campylobacter was the only pathogen detected by GMP/culture and what was the number of co-infections between Campylobacter and another enteropathogen?

The two GMP used can only detect C. jejuni/C. coli? Or can they also detect other Campylobacter species? Could this be one of the reasons for the difference in sensitivity of GMP versus culture?

Line 72: Please indicate the method used to generate a microaerobic atmosphere, the temperature, and the incubation time of the plates.

Fig. 1: The scientific names of the bacterial genera are misspelled

Reviewer 3 Report

The authors present study on prevalence of Campylobacter species in humans from private non-profit hospital in Santiago, Chile. Although the results are of limited scientific importance and novelty, their data helps to complete general knowledge of the most commonly reported gastrointestinal bacterial pathogen in the world and will be probably more interesting to readers from South America.

I think that the whole manuscript should be revised by a native speaker in order to improve English language.

Title

I find title a bit misleading, as there are much more pathogens involved in the study (Salmonella spp., Shigella spp. etc. ). I understand that the authors got all of these results using GMP, but I think they should either remove it from the manuscript or put it in Title and/or keywords. Also, I assume that retrospective  in title means that these are not fresh data because samples are from 2014-19. I think it is double information and I would remove either retrospective or time frame from the title. Also, I assume that with monocentric authors wanted to stress that samples are from one city only. I think that the title would have same information without retrospective  and monocentric.

Abstract

In Abstract and later throughout the manuscript authors write Salmonella sp. I don't understand this, as other genera they write spp. If they mean only S. enterica, than they should write it, and not Salmonellla sp.

Materials and Methods

Authors should add a paragraph describing samples used in the study.

line 71: Culturing instead of Culture

line 74: were instead of was.

Results

line 113: How did the authors identified these 4533 specimens? Randomly? Please explain.

Discussion

line 131: Can you explain what are routine samples? I assume these are stool samples from persons with diarrhea ..

line 154: I cannot find paper Castillo et al in References

Reviewer 4 Report

The article “Campylobacter spp. prevalence in Chile: a retrospective monocentric study in Santiago based on molecular detection in clinical stool samples from 2014 to 2019” aims to investigates the prevalence of enteropathogenic samples among patients using Gastrointestinal multiplex PCR panels. The study is well designed and interesting as it shows Campylobacter spp. is the most prevalent enteropathogen in Chili. However, there are some minor points which need to be addressed

1.     Introduction need to emphasis on the prevalence of Campylobacter spp. in Chili and other South American countries.

2.     What as the reference strains used?

3.     Please describe the nature of sample tested (stool samples, or isolates) and their clinical relevance?

4.     What is the specificity and the sensitivity of Gastrointestinal multiplex PCR panels?

5.     Add the P-values to signify the statistical differences in the result section.

6.     The discussion needs to be concise.
